# Interaction between Health and Financial Status on Coping Behaviors during the COVID-19 Pandemic

**DOI:** 10.3390/ijerph192013498

**Published:** 2022-10-19

**Authors:** Mehmet Yanit, Kan Shi, Fang Wan, Fei Gao

**Affiliations:** 1Asper School of Business, University of Manitoba, Winnipeg, MB R3T 5V4, Canada; 2The Institute of Wenzhou Development Model Research, Wenzhou University, Wenzhou 325035, China; 3Marketing Department, Bentley University, Waltham, MA 02452, USA

**Keywords:** financial status, shift-and-persist, COVID-19, pandemic, confidence

## Abstract

Background: The year 2022 started with protests against COVID-19 restrictions throughout North America. These events manifest the fact that some segments of the population are not compliant with the preventive measures of COVID-19, and the reasons of the disobedience against public health regulation remain unclear. The current paper examined the joint effect of financial and health status on people’s likelihood of pursuing active coping efforts (i.e., following preventive measures) and giving up coping with the COVID-19 pandemic. Method: We conducted a large-scale survey study in China (*N* = 3834) in May 2020. Results: Our results showed that people with low financial status were less likely to manifest active coping behavior and more likely to give up coping with the pandemic. People’s self-confidence in coping with the pandemic mediated this effect. We showed that one’s health status could interact with their financial status in a way that healthy people with low financial status would have less confidence in their coping abilities and thus become less likely to pursue active coping efforts and more likely to give up coping with the pandemic. Conclusions: Our results call for policymakers to find more effective solutions for noncompliant groups so that they can abide by the general guidelines in the COVID-19 context and other social crises that may emerge in the future. We suggest that governments should concentrate their support efforts on healthy populations of low financial segments to prevent COVID-19 and other infectious diseases in the future from spreading further.

## 1. Introduction

COVID-19 is one of the most contagious viruses that humankind has ever faced. As of the writing of this article (7 February 2022), more than 406 million people had been infected, and 2.2 million people had lost their lives [1]. Even though many countries had successfully vaccinated most of their populations, they keep asserting preventive policies for COVID-19, such as wearing a mask in public places, practicing social distancing, and limiting working capacity. Some people rigidly observe these protocols, whereas others relinquish them [2] or deem them unnecessary [3] by eventually posing significant risks to public health [4]. This attitude toward COVID-19 restrictions had recently turned into public action, as displayed in public protests against restrictions. For instance, starting as small-scale local demonstrations, the protests against COVID-19 preventive measures had reached their peak among the Canadian public in the first quarter of 2022, evolving into a nationwide protest with thousands of Canadians demonstrating their opposition to ongoing protocols [5]. The United States also witnessed ongoing protests, with thousands of truckers gathering on the outskirts of Washington, DC, to force authorities to remove the restrictions [6]. In China, ongoing restrictions generated resistance among some segments of the public. For example, in their interview-based study with elders in China, ref. [7] showed that Chinese elders largely object to the public intervention and claimed that the interventions limit their mobility and reduce the quality of their life.

The objectors’ behaviors motivated our first research question. Why do some people give up or refuse to collaborate in the public’s efforts to control the spread of the virus and restore public life? In the current research, we attempted to answer this question and identify these segments by looking at the situation through the lenses of self-assessed health and general financial status of people and examining the joint effect of one’s financial and health status on their confidence in coping with the pandemic and, in turn, on their active coping and giving-up behaviors in the efforts against COVID-19.

In dealing with health-related adversities, individual differences (i.e., personality and financial status) can create significant behavioral variances. In the context of the COVID-19 pandemic, for instance, prior work examined people’s personality traits and health behavior [8,9]. In this line of work, ref. [10] showed that people’s adherence to preventive policies during the COVID-19 pandemic is affected by extroversion, showing that extrovert individuals were more likely to relinquish mobility restrictions during the pandemic. The author of [11] found that people with dark personality traits were less likely to follow health protocols and be influenced by compassion-focused public messages to protect vulnerable segments during the pandemic. More relevantly, ref. [12] demonstrated that COVID-19 cases were more prevalent in low socioeconomic neighborhoods than in high socioeconomic ones. They theorized that people from low financial segments have less ability to abide by protective measures because they are more likely to be essential workers who cannot remotely work [12] and less likely to follow safety protocols as most of them also have poor health literacy [13].

Prior research examined the effects of individuals’ financial status on their coping behavior under health-related stressors. This line of work concluded that people of low financial status manifest a “shift-and-persist” strategy by regulating their emotions with a self-fabricated optimism (shift) and adapting themselves to live with the threat (persist) [14,15,16] instead of making any effort to eliminate the threat [16], because they lack the required financial resources to back up their efforts even if they make any [17].

Indeed, research, to date, has consistently shown that the “shift-and-persist” model is more pervasive among individuals of low socioeconomic status [14]. We argue that the “shift-and-persist” strategy allows individuals with few financial resources to create an illusionary optimism that serves as a protective and defensive mechanism. For instance, ref. [18] showed that “shift-and-persist” serves as a protective mechanism among youth with low financial status who struggle with a high body mass index (BMI). The authors of [19] found that the application of shift-and-persist alleviates the psychosocial stress for individuals of low financial status. More recently, ref. [20] reported that Mexican American youth from low socioeconomic backgrounds successfully protected themselves against depressive symptoms caused by peer discrimination by applying shift-and-persist strategies. Therefore, accepting life stressors without making an active effort to eliminate them (shift-and-persist) is a frequently adopted strategy by people with low financial status. This passive coping strategy is imprinted in these individuals’ life journeys starting from their early years as means to survive adversities [15].

Based on these findings, we propose that due to greater (vs. lower) compatibility between low (vs. high) financial status and the shift-and-persist model [14,16,17], people with low financial status are more inclined toward accepting the threat during the COVID-19 pandemic, more reluctant to make active coping efforts (i.e., less active coping behavior), and more likely to give up any efforts to cope with the pandemic. These suggestions bring a second research question under our scope. Why are people from low financial status more likely to apply the shift-and-persist strategy during a pandemic?

Prior literature suggested that the differences in financial status can create discrepancies in people’s confidence in dealing with life stressors [21,22]. Confidence refers to a state of mind that directly impacts one’s perception of dealing with a problem [23]. Prior research also showed that people with greater confidence show more active coping and less give-up behaviors than people with lower confidence [24]. Thus, we propose that people with higher (vs. lower) financial status are likely to have greater (vs. less) confidence in coping with COVID-19 and thus are more likely to show active coping (vs. give-up) in the pandemic compared with those with lower (higher) financial status.

Prior research showed that the accessibility to supportive resources can generally affect one’s confidence in dealing with a problem [25]. In this case, one’s resources include both *tangible* resources such as money or financial assets [26] and *intangible* resources such as one’s physical health [27]. We observed that whereas prior work mainly focused on how tangible resources such as financial status impact one’s confidence in dealing with health-related problems, very few studies explored how intangible resources such as health status and tangible resources such as financial status interactively influence one’s confidence in dealing with health-related problems and subsequent coping behavior. In particular, for individuals with relatively low financial status, their health status is likely to play an important role in specifying their confidence in coping abilities and subsequent behavior. We propose that for healthy individuals who have fewer financial resources (i.e., lower financial status), their health becomes an asset at risk during the pandemic. Since they do not have access to the financial resources that can sufficiently mitigate or remove risks associated with their health, those individuals are likely to experience a confidence deficiency in their coping abilities [21,28]. Past research showed that a lack of confidence in one’s abilities to attain certain goals can undermine their commitment to the goal [29,30].

Building on this premise, we argue that compared with individuals who have high financial resources (regardless of their health status), individuals who have low financial resources, yet a high health status, are more likely to accept the COVID-19 threat in their lives as it is instead of making an effort to eliminate it or prevent it from spreading further. This is a typical manifestation of the shift-and-persist model discussed earlier, as this behavioral schema is an essential survival mechanism to remain healthy and should be more extensively pursued by healthy individuals under the pressure of low financial status. In other words, as profiled in the antirestriction protestors, we expect healthy individuals with low income to be more reluctant to abide by COVID-19 restrictions (less active coping) or to completely relinquish the restrictions (giving up) due to their lack of self-confidence in coping with the COVID-19 pandemic. Therefore, we propose that healthy but financially poor individuals have weaker commitments to their efforts to cope with COVID-19 compared with unhealthy and financially poor individuals. However, we propose that this effect is muted for people with high financial status. High financial status serves as a buffer against risks associated with one’s health [31,32], since financial status directly determines one’s accessibility to medical resources. In other words, people with high financial status do not need to pursue shift-and-persist to protect themselves as they already possess the resources to fight against COVID-19. This situation makes people with high financial status more likely to employ active coping strategies and less likely to give up regardless of their health status.

In the present study, we conducted a large-scale survey in China to explore how individuals’ health and financial status jointly affects their confidence in coping abilities with COVID-19 and their subsequent active coping and giving-up behaviors. At the time of data collection, the restrictions implemented by the Chinese authorities included village isolations [33], travel restrictions [34], community lockdowns [35], public transport shutdowns [36], and school closures and remote teaching [37]. Therefore, we tested our research questions in an environment where the effect of restrictions on the public was highly pronounced. In summary, our research questions were as follows:Why do some people give up or refuse to collaborate in the public’s efforts to control the spread of the virus and restore public life?Why are people from low financial status more likely to apply the shift-and-persist strategy during a pandemic?

## 2. Method

### 2.1. Sample and Recruitment

During the peak period of the COVID-19 outbreak, in May 2020, we recruited 3834 Chinese participants (55% women, 50% aged 30 and over) from mainland China via a local online participant panel (So Jump). Although the participants were majorly from Chinese cultural backgrounds, the results of this study can still be applied in western cultural contexts for the following reasons. Firstly, the new Chinese generation largely believe in the same universal values as western populations such as human rights, democracy, and freedom [38]. Therefore, Chinese people are likely to manifest similar needs and expectations to their western counterparts during the pandemic. Secondly, despite the cultural heterogeneity across different countries, the types of public interventions applied by the authorities [39,40] and their social and economic impacts on the society are quite homogenous [40]. Lastly, research showed that objections to COVID-19 restrictions in China [7] are part of the global phenomenon in which certain public segments oppose the active coping and intervention measures during the COVID-19 pandemic.

The G*Power analyses [41] showed that recruiting 3834 participants would let us detect a small effect size (f ^2^ = 0.02) with 99% power when type 1 and type 2 error rates are assumed to be equally important for our analyses. No participant data were excluded. Each participant received the equivalent of USD1.5 in Chinese Yuan for their participation. This study was approved by the research ethics review committee of Wenzhou University, China. The data, R code, and survey are available at https://doi.org/10.6084/m9.figshare.19702021.v1, accessed on 21 September 2021.

### 2.2. Measures

#### 2.2.1. Active Coping

We measured active coping with two items by asking the participants to rate the degree to which they agreed or disagreed that they have been concentrating their efforts on doing something about the COVID-19 situation or they have been taking action to make the situation better (1 = strongly disagreed, 4 = strongly agreed). The items were adapted from the Coping Orientation to Problems Experienced (COPE) scale [42]. The correlation between these scale items was 0.75, indicating high in-scale reliability.

#### 2.2.2. Give-Up

The give-up scale consisted of two items adapted from the COPE scale [42]. The participants were asked to rate the degree to which they agreed or disagreed that they have given up trying to deal with the COVID-19 situation or they have given up the attempt to cope on a 4-point Likert scale. The correlation coefficient between these scale items was 0.87, indicating high in-scale reliability.

#### 2.2.3. Confidence in Coping Abilities

To measure confidence in coping abilities, we asked the participants to respond to four items using a 4-point Likert scale, rating the degree to which they agreed or disagreed with the following statements: (a) “I am confident that I can deal with the COVID-19 situation efficiently”; (b) "Thanks to my resourcefulness, I am confident on how to handle the situation"; (c) “I can remain calm when facing this difficulty because I am confident that I can rely on my coping abilities”; and (d) “When I am confronted with this problem, I am confident that I can usually find several solutions.” These items were adapted from Problem Solving Inventory (PSI)—Form A developed by [43]. Moreover, reliability analyses with these in-scale items manifested a high Cronbach’s alpha value of 0.90, which implies high in-scale reliability.

#### 2.2.4. Optimism

We measured the participants’ optimism with two items adapted from the COPE scale [42]. The participants rated the degree to which they agreed or disagreed with the statements “I have been trying to see in a different light, to make the situation seem more positive” and “I have been looking for something good in what is happening” on a 4-point scale. Correlation analyses showed that the correlation coefficient between these scale items was 0.62, which indicates high in-scale reliability.

#### 2.2.5. Denial

To measure denial, the participants rated the degree to which they agreed or disagreed with the statements “I have been saying to myself ‘this is not real’” and “I have been refusing to believe that it has happened” on a 4-point scale. These two items were also adapted from Carver’s COPE scale [42]. The correlation coefficient between these scale items was 0.79. Therefore, we concluded that the in-scale reliability was sufficiently high.

#### 2.2.6. Anxiousness

We measured anxiousness by asking the participants to rate the degree to which they agreed that they felt they were anxious all the time on a 5-point scale (1 = Not at all, 5 = Extremely) [44].

#### 2.2.7. Health Status

To measure people’s health statuses, we asked the participants to rate how healthy they were in general on a 5-point scale (1 = *Poor*, 5 = *Excellent*). For brevity, this variable will be referred to as “Health” for the rest of this paper.

#### 2.2.8. Financial Status

This research used poverty measures to measure one’s financial status. Combining poverty scales developed by [45,46], we measured the participants’ financial status using a 7-point Likert scale with a total of 9 reverse coded and standard items such as “In the past year, there were some days when I or someone from my family went hungry because we did not have enough money for food”, “I have difficulty to find a balance between my income and expenses at the end of the month”, or “Compared to other people living in my country, my current living standards are better”. In addition, in-scale reliability was sufficiently high with Cronbach’s alpha value of 0.78.

### 2.3. Procedure

The participants answered questionnaires in the following order: Active Coping, Give-Up, Confidence in Coping Abilities, Optimism, Denial, Anxiousness, Health, and Financial Status. The questionnaire ended with demographic questions (including sex, age, education level, and marital status). The study took approximately 15 min to complete. Scale items are provided in the Appendix A.

### 2.4. Statistical Analyses

The data were analyzed using various R packages and Hayes’s PROCESS module on SPSS [47]. To analyze the main effect of financial status on active coping and give-up behaviors along with the suggested covariates, we conducted OLS regressions with *lm* function in the R environment [48]. To analyze the mediation and moderated mediation paths, we used model 7 from Hayes’s PROCESS module on SPSS [47].

## 3. Results

We tested the main effect of financial status on active coping and give-up behaviors in two separate models. We adopted a stepwise approach as in [49], in which the first step consisted of demographic variables such as age, sex, marital status, education level, and other control variables such as optimism, denial, anxiousness, and health status. In the second step, financial status was added. We aimed to show the unique variance generated by financial status on active coping and give-up behaviors.

In mediation and moderated mediation analyses, we adopted similar approaches. In the mediation analyses, we checked the direct and indirect effects of financial status on active coping and give-up behaviors in two models. In the moderated mediation analyses, we examined the effect of the interaction between financial status and health status on confidence in coping abilities and, in turn, on active coping and give-up, respectively. The suggested models have also been simultaneously controlled for covariate variables.

Descriptive statistics are presented in Table 1. The correlation matrix of variables of interest is presented in Table 2.

### 3.1. Effect of Financial Status on Active Coping and Give-Up

Results of OLS regressions showed that financial status positively predicted people’s active coping tendencies (β = 0.05, *SE* = 0.01, *p* < 0.001) while negatively predicted giving up (β = −0.08, *SE* = 0.01, *p* < 0.001; see step 2 on Table 3) after controlling the demographic variables such as age, sex, marital status, and education level and other variables of interests such as optimism, denial, anxiousness, and health status. Furthermore, a model comparison between the regression steps without (step 1) and with financial status as an additional predictor (step 2) for both models 1 and 2 showed that financial status significantly explains unique variance in both active coping (RSS_1_ = 1721.1, RSS_2_ = 1713.4, F-score = 17.13, *p* < 0.001) and give-up values (RSS_1_ = 1788.3, RSS_2_ = 1767.8, F-score = 44.22, *p* < 0.001).

### 3.2. Mediating Effect of Confidence in Coping Abilities

We next examined whether the relationship between financial status and active coping or give-up was mediated by confidence in coping abilities on Hayes’s PROCESS module on SPSS [47]. After controlling for the demographic variables (age, sex, marital status, and education level) and other variables of interests (optimism, denial, anxiousness, and health status), the mediation analyses showed that confidence in coping abilities indeed mediated the relationship between financial status and active coping (see Figure 1a). The indirect effect of financial status on active coping was positive and significant (β = 0.02, 95% CI = [0.01, 0.02]). The relationship between financial status and giving-up tendencies was also mediated by confidence in coping abilities (see Figure 1b). The indirect effect of financial status on give-up through confidence in coping abilities was negative and significant (β = −0.01, 95% CI = [−0.01, −0.003]).

### 3.3. Moderating Effect of Health Status

Finally, we examined whether the mediation effect of confidence in coping abilities for the relationship between financial status and active coping or giving-up tendencies is moderated by health status using PROCESS on SPSS [47]. We specifically examined the interaction effect between financial status and health status on active coping and give-up through confidence in coping abilities (see Figure 2a,b) as the model controlled by the same covariates we tested in the previous analyses.

The results revealed a significant interaction effect between financial status and health status on confidence in coping abilities (β = 0.03, *SE* = 0.01, *p* < 0.05) (see Figure 3). After controlling for demographics and other variables of interests, further analyses revealed significant indirect effects of interaction in each level of health status through confidence on active coping (for low health: β = 0.01, *SE* = 0.003, CI = [0.01, 0.02]; for medium health: β = 0.01, *SE* = 0.003, CI = [0.01, 0.02]; for high health: β = 0.02, *SE* = 0.003, CI = [0.01, 0.02]) and give-up (for low health: β = −0.004, *SE* = 0.002, CI = [−0.01, −0.002]; for medium health: β = −0.01, *SE* = 0.002, CI = [−0.01, −0.003]; for high health: β = −0.01, *SE* = 0.002, CI = [−0.01, −0.004]).

More importantly, significant indices of moderated mediation in both model 1 (β = 0.01, 95% CI = [0.001, 0.01]) and model 2 (β = −0.002, 95% CI = [−0.004, −0.0003]) suggested that the indirect effect of financial status on active coping and give-up was significantly larger for healthy people (high health) than people with moderate and low health. That said, healthy people with low financial status had significantly less confidence in their coping abilities, and, in turn, they became less likely to employ active coping behavior and more likely to give up compared with other segments.

### 3.4. Robustness Check

To further test the robustness of the proposed moderated mediation models, we have tested potential alternative mediators to confidence in coping abilities in the focal moderated mediation models. The covariate analyses in the initial models showed that, akin to confidence, optimism and denial also had significant effects on both dependent variables (see Table 4). Therefore, we have tested those variables as alternative mediators in our models while simultaneously controlling for the remaining variables in each competing model.

The results showed that when separately tested, for each potential mediator, confidence intervals of the moderated mediation indices intersected with zero (see Table 5). This result suggests that none of the potential mediators could significantly mediate the moderated mediation models at the 95% confidence level. Hence, we could rule out the alternative explanations to the suggested moderated mediation models, further contributing to the robustness of confidence in coping abilities as a mediator in this focal relationship.

In this study, we found support for our propositions. We showed that financial status could predict one’s likelihood of employing active coping methods and giving-up tendencies in coping with COVID-19. People with low financial status were less likely to actively cope with the pandemic and more likely to give up coping with it. This effect remained significant when it was tested with covariates. Moreover, we showed that these effects are mediated by one’s confidence in their coping abilities in a way that low financial status people showed less confidence in their coping abilities, and, in turn, they became less likely to actively cope with the pandemic and more likely to give up coping with it. Finally, our moderated mediation analyses with participants’ self-reported health status showed that healthy people with low financial status were significantly more likely to experience confidence deficiency in their coping abilities, and, in turn, they were less likely to employ active coping and more likely to give up coping with the pandemic. The general discussion section discusses the theoretical and practical implications of these findings.

## 4. General Discussion

In the current research, we attempted to identify noncompliant segments with COVID-19 restrictions via their self-reported health and financial status by examining the combined effect of these factors on one’s active coping behavior and giving-up tendencies. The results showed that low-financial-status people are more likely to give up coping with COVID-19 restrictions. Within the low financial segment, healthy individuals showed less willingness to employ active coping and were more likely to give up coping with the pandemic. These results suggest several theoretical and practical implications.

Firstly, our work enriches existing literature on the behavioral variance of the public during the pandemic. Prior work explored the effect of sex [50], cultural values [51], and ideological differences of individuals [52] on their COVID-19-coping behaviors. These studies showed that men were more likely to relinquish COVID-19 restrictions, so were people from loose cultures and those who strongly identified with their political party and ideologies. Therefore, the current paper explored understudied drivers of public health behaviors, such as financial and health status. Our results imply that public health officials should consider the lack of financial resources and institutional support [53] as factors that make people abandon active coping efforts while making them more likely to give up coping. We showed that people’s reduced active coping behavior and increased giving-up tendencies are mainly driven by their lack of financial resources to cope with the crisis. Hence, public health officials may use economic incentives to boost the measures against COVID-19 for those in need of financial support to efficiently follow mandates. For example, a recent news report from China showed that to incentivize older people to take COVID-19 vaccination, the government gives them shopping coupons, free groceries, or even cash rewards [54].

Secondly, vulnerable groups such as minorities, refugees, and blue-collar workers are already exposed to a greater risk than the rest of the population during the pandemic [55,56]. These groups mostly live in condensed and overcrowded communities [57], facilitating the spread of the virus. For example, research showed that during the COVID-19 restrictions, rural-to-urban migrants in urbanizing China lived in an overcrowded environment and demonstrated a greater risk of spreading the virus, but are the least likely to report COVID-19 symptoms to avoid discriminatory treatment [58]. By showing that low-financial-status groups are more likely to give up, our results suggest that healthy individuals of these segments may inadvertently pose a threat to more vulnerable groups within their communities, such as elders and those with chronic illnesses. These healthy individuals have greater mobility and a greater likelihood of spreading the virus in and out of their community because of the type of jobs they hold [59]. Our research provided some evidence to support this idea. However, further research is needed to identify the segments who spread the virus more actively so that governments’ financial relief efforts can more precisely focus on these risky segments and keep them away from physical work environments and subsequent risky behaviors [60].

Thirdly, antagonists to restrictions tend to deem them unnecessary with an overall “If I can adapt, you can adapt, too” attitude toward the supporters of COVID-19 restrictions [3]. Indeed, the latest protests even witnessed various fights between pro- and antirestriction groups, with some escalating to sheer violence [61]. As implied by our results, if such an attitude is specific to a certain social class, in the long run, we can expect current conflicts over restrictions to result in greater polarization and marginalization between socioeconomic classes in society [62]. Therefore, we caution the policymakers to take necessary measures to prevent current public tension from growing further.

Fourthly, our study employed a Chinese population from diverse occupational backgrounds in the early periods of the COVID-19 pandemic when the severity of the pandemic was at its peak in China. These people were among the most affected by the pandemic since they experienced the virus on ground zero during the pandemic’s heydays. Despite being under significant risk and pressure, our results showed that many Chinese participants were inclined to give up. Considering the collectivist structure of Chinese culture [63,64] with a public emphasis on the community’s wellbeing [65], our research results indicate that even in highly collectivist cultures, people are inclined to give up and dismiss the potential risks that their behaviors can pose to others. Theoretically, this poses an interesting research area for the future. When it comes to coping with adversity with limited resources (such as the individuals with low financial status as in our sample), their lack of confidence led them to give up coping behaviors against COVID-19 with the cost of relinquishing their collectivistic ideals and the cultural norm that places the consideration of other people as priority.

Finally, policymakers and public health officials employed punitive measures to introduce behavior change to violators of restrictions since the beginning of the pandemic. Similarly, media reporters employed shame tactics to pressurize these segments into changing their behaviors [66]. However, our research suggested that directly addressing the financial resources of people with low financial status is more effective and more potent than existing tactics. Public officials can address financial disparities by enhancing individuals’ access to health-related services and products during the COVID-19 pandemic. Consequently, further research should examine what effective communications strategies should be used by public officials to increase the self-confidence of vulnerable groups in fighting against the pandemic.

## 5. Limitations

Our results are not without limitations. For example, our results in this research are limited to a Chinese population. In order to control for potential variances that can emerge in the observed mechanism due to cultural differences, the suggested conceptual framework should be tested in a different cultural context.

On the other hand, considering that our study was conducted in the early period of COVID-19 in the center of the pandemic, high mortality salience may have played a role in the relationship between financial status and give-up tendencies in our analyses. The authors of [67] argued that increased mortality salience during the onset of the COVID-19 pandemic could cultivate greater conservatism and reduce people’s openness to changing their lifestyles. Therefore, further research needs to examine the effect of mortality salience on low-financial-status people’s health behaviors.

More importantly, the ongoing misinformation crisis about COVID-19 may have a latent effect on the conceptual relationships we observed in this research. The authors of [68] showed that many people consider the COVID-19 pandemic no more dangerous than seasonal flu. Misinformation in the COVID-19 context is particularly dangerous as it may lead people to underestimate the severity of the pandemic and become more likely to abandon protective measures [69] or follow conspiracy theories, which can further lower their trust in the authorities and policymakers [70]. Low-financial-status people, due to their low health literacy, may be more predisposed to such information contamination and, therefore, more likely to manifest the behaviors that we observed in the current research. Future research needs to study the role of misinformation in stripping people off their guard against the pandemic. Hence, verification and authentication of pandemic-related information would be another avenue for policymakers.

## 6. Conclusions

There is a consensus that preventive measures have effectively prevented infection rates from soaring [71]. Nevertheless, the efficacy of preventive measures depends on people’s willingness to follow them [72]. This research attempted to identify the social segments that are most likely to abandon preventive measures based on their health and financial status. Our results call for policymakers to find more effective solutions for noncompliant groups so that they can abide by the general guidelines in the COVID-19 context and other social crises that may emerge in the future.

## Figures and Tables

**Figure 1 ijerph-19-13498-f001:**
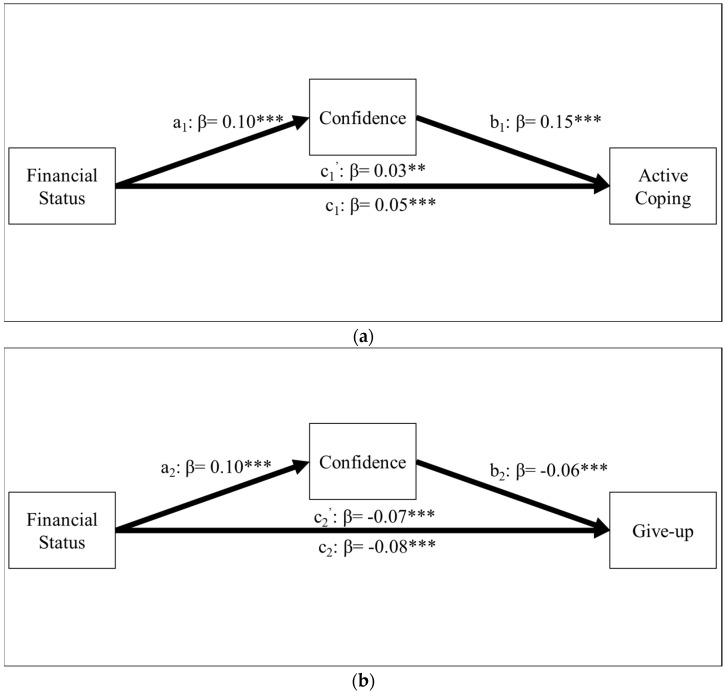
(**a**) Confidence in coping abilities mediates the relationship between financial status and active coping (model 1). (**b**) Confidence in coping abilities mediates the relationship between financial status and giving-up tendencies (model 2). ** *p* < 0.01. *** *p* < 0.001.

**Figure 2 ijerph-19-13498-f002:**
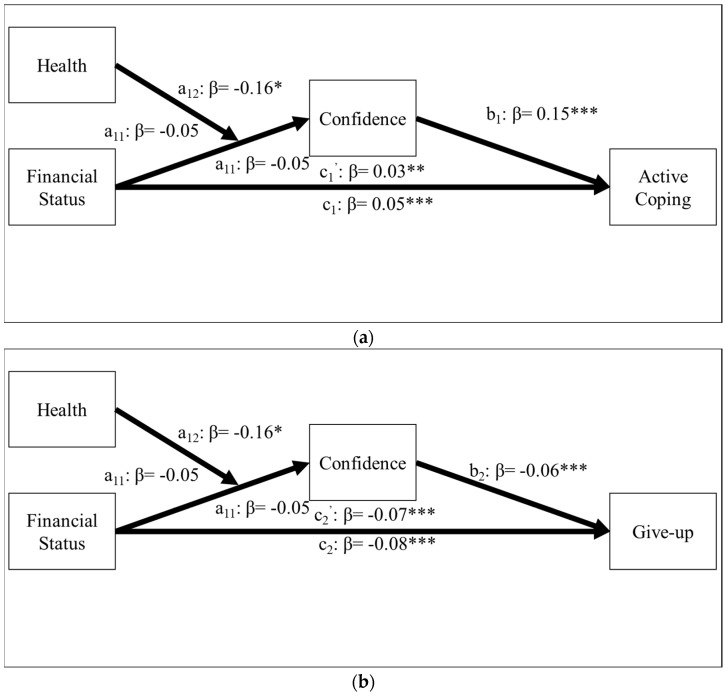
(**a**) Moderated mediation effect on active coping. (**b**) Moderated mediation effect on giving-up tendencies. Parameter estimates for moderated mediation analysis using healthiness as potential moderator after controlling for covariates. As ‘i’ denotes model number, a_i1,_ a_i2_, and a_i3_ indicate the direct effect of financial status on confidence, direct effect of health on confidence, and interaction effect between financial status and health on confidence, respectively, in both models. * *p* < 0.05. ** *p* < 0.01. *** *p* < 0.001.

**Figure 3 ijerph-19-13498-f003:**
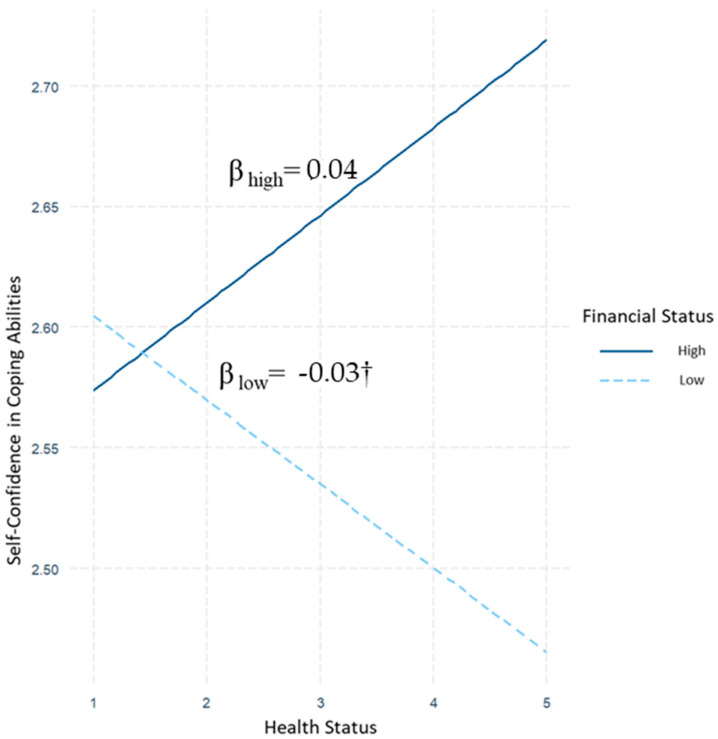
Graph of interaction between financial status and health on confidence in coping abilities. Note. † *p* < 0.10.

**Table 1 ijerph-19-13498-t001:** Descriptive Statistics (*N* = 3834; 55% Women, 50% aged 30 and over).

Variable	*M*	*SD*	Reliability
Active	2.49	0.86	*r* = 0.75
Give-up	1.49	0.82	*r* = 0.87
Confidence	2.59	0.84	α = 0.90
Optimism	2.39	0.85	*r* = 0.62
Denial	1.74	0.88	*r* = 0.79
Anxiousness	2.26	1.21	N/A
Health	4.36	0.84	N/A
Financial Status	4.70	1.04	α = 0.78

Note: Reliability has not been calculated for single-item scales.

**Table 2 ijerph-19-13498-t002:** Correlation Matrix of Variables of Interest.

Variable	1	2	3	4	5	6	7	8
Active	-							
Give-up	0.18 ***	-						
Confidence	0.33 ***	0.02	-					
Optimism	0.59 ***	0.32 ***	0.32 ***	-				
Denial	0.36 ***	0.53 ***	0.09 ***	0.33 ***	-			
Anxiousness	–0.02	0.11 ***	–0.16 ***	–0.03	0.12 ***	-		
Health	–0.04 *	–0.10 ***	–0.01	–0.07 ***	–0.15 ***	–0.07 ***	-	
Financial Status	0.07 ***	–0.18 ***	0.18 ***	0.07 ***	–0.23 ***	–0.19 ***	0.04 *	-

Note: * *p* < 0.05. *** *p* < 0.001.

**Table 3 ijerph-19-13498-t003:** Stepwise Regression Results and Model Fit Metrics for Main Effects.

Step and Variable	Model 1: Active Coping	Model 2: Give-Up
*Step 1*		
Age	–0.02 (0.01) †	–0.01 (0.01)
Sex (Men)	0.02 (0.02)	0.04 (0.02) †
Marital Status (Married)	–0.04 (0.03)	–0.05 (0.03) †
Education Level	0.05 (0.01) ***	–0.03 (0.01) **
Optimism	0.52 (0.01) ***	0.17 (0.01) ***
Denial	0.19 (0.01) ***	0.42 (0.01) ***
Anxiousness	–0.02 (0.01) *	0.04 (0.01) ***
Health	0.02 (0.01)	–0.05 (0.01) ***
*Adjusted R-squared*	*0.3852*	*0.3080*
*Step 2*		
Age	–0.03 (0.01) *	0.002 (0.01)
Sex (Men)	0.03 (0.02)	0.03 (0.02)
Marital Status (Married)	–0.04 (0.03)	–0.05 (0.03) †
Education	0.04 (0.01) ***	–0.01 (0.01)
Optimism	0.52 (0.01) ***	0.18 (0.01) ***
Denial	0.20 (0.01) ***	0.41 (0.01) ***
Anxiousness	–0.01 (0.01)	0.02 (0.01) **
Health	0.01 (0.01)	–0.04 (0.01) **
Financial Status	0.05 (0.01) ***	–0.08 (0.01) ***
*Adjusted R-squared*	*0.3878*	*0.3158*
*Δ R-squared*	*0.0026 ****	*0.0078 ****
*F-test*	*17.13*	*44.22*

Note: The table shows standardized regression coefficients, with standard errors in parentheses and adjusted R-squared values for each step to compare model fits between steps. † *p* < 0.1. * *p* < 0.05. ** *p* < 0.01. *** *p* < 0.001.

**Table 4 ijerph-19-13498-t004:** Coefficient parameters for variables in moderated mediation models after controlling for covariates.

Variables	Model 1: Active Coping	Model 2: Give-Up
Age	–0.03 *	0.004
	(0.01)	(0.01)
Sex (Men)	0.01	0.04 †
	(0.02)	(0.02)
Marital Status (Not married)	–0.05 †	–0.05 †
	(0.03)	(0.03)
Education	0.03 **	–0.002
	(0.01)	(0.01)
Confidence	0.15 ***	–0.06 ***
	(0.01)	(0.01)
Optimism	0.47 ***	0.20 ***
	(0.01)	(0.01)
Denial	0.19 ***	0.41 ***
	(0.01)	(0.01)
Anxiousness	–0.004	0.02 *
	(0.01)	(0.01)
Financial Status	0.03 **	–0.07 ***
	(0.01)	(0.01)
Index of Mod. Med.	[0.001, 0.01]	[–0.004, –0.0003]

Note: † *p* < 0.10. * *p* < 0.05. ** *p* < 0.01. *** *p* < 0.001.

**Table 5 ijerph-19-13498-t005:** Moderated mediation indices with different mediator candidates.

Potential Mediators	Active C	Give-Up
	95% CI	95% CI
Optimism	[–0.01, 0.02]	[–0.003, 0.01]
Denial	[–0.003, 0.01]	[–0.01, 0.02]

## Data Availability

The data, R code, and survey items used in this paper are available at https://doi.org/10.6084/m9.figshare.19702021.v2 (accessed on 21 September 2021).

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
