# Peer review of "Interaction between Health and Financial Status on Coping Behaviors during the COVID-19 Pandemic"

_ijerph, 2022, doi:10.3390/ijerph192013498_

Round 1

Reviewer 1 Report

Dear Authors,

I would like to thank you for the opportunity to see this very interesting work. At the beginning I would like to emphasize that the size of the sample collected by you makes an amazing impression. However, I would like to suggest a few changes that, in my opinion, would improve the manuscript you prepared.
- in the first place, I would like to suggest that in the beginning of the text we should refer additionally to the situation in China - there would be data collected and a description of the situation in this country would be valuable information for the reader.
- you write about the situation from the first quarter of 2022 (in Canada), which contributed to the formulation of the first research question - the study was conducted according to the information in the abstract in 2020, i.e. almost two years earlier; during this time the situation could change
- I would suggest that at the end of the introduction, e.g. research questions be presented in points - it will help to summarize the content you presented
- I suggest describing the situation at the time of data collection in a bit more detail - possibly what were the restrictions, etc.
- When describing the tools, information about their reliability should be added
- I would suggest moving the beginning of the "Results" section to Methods and describing it as, for example, "Statistical analysis".
- I suggest adding information about the sample structure to table 1 - N /% women / men, average age, etc.
- Maybe it is just my imagination, but Tables 4 and 5 need to be reformatted because they look like Figures

Reviewer 2 Report

Thanks for inviting me to review this manuscript. This paper investigates the interaction between health and financial status on (active/give-up) covid coping behaviours based on a relatively large scale online survey in China. In general, this is very interesting but it still can be improved in the following ways:

Major comments:

1.     My major concern is that, in the introduction section, it seems that this study aims to understand why some people protest against Covid containment interventions in North American contexts but it uses data from a very different context, China for the analysis. I understand maybe it is due to the availability of data but the author needs to explain why empirical results from China can be transferred to North American contexts at the beginning of the method section (section 2.1).

I think the following three points may be helpful.

First, although there are cultural differences between China and North America, people share very similar fundamental needs and values (see e.g., Wang, 2012). Second, Covid containment interventions have diffused across countries (e.g., Lee et al., 2021; Sebhatu et al., 2020) and consequently people may react to them in a similar way. Third, although public protests against Covid-19 preventive measures have never been reported in China, previous studies revealed that some people intensely opposed most of the Covid-19 policies, even when Covid containment interventions were most strictly enforced (Liu et al., 2021).

2.     In the discussion section (section 4), it’s very good to see some discussion about the vulnerable group. I think it is of special interest to talk about rural-urban migrants during the pandemic. They are a very iconic vulnerable group in urbanising China. Having low financial status generally, they also suffered from various difficulties such as discrimination, forced eviction, and environmental racism. In order to avoid such difficulties, many of them concealed reporting symptoms that may be considered as Covid-19 (Liu et al., 2022). This may also be useful to explain why people with lower financial status are more likely to give up Covid-coping.

Minor comments:

1.     Fig. 1 a, b and Fig. 2 a, b are ugly. Please make them smaller and use larger font size.

Reference

Lee, J. K., Bullen, C., Ben Amor, Y., Bush, S. R., Colombo, F., Gaviria, A., ... & Lancet COVID-19 Commission Task Force for Public Health Measures to Suppress the Pandemic). (2021). Institutional and behaviour-change interventions to support COVID-19 public health measures: a review by the Lancet Commission Task Force on public health measures to suppress the pandemic. International health, 13(5), 399-409.

Liu, Q., Liu, Y., Zhang, C., An, Z., & Zhao, P. (2021). Elderly mobility during the COVID-19 pandemic: A qualitative exploration in Kunming, China. Journal of transport geography, 96, 103176.

Liu, Q., Liu, Z., Kang, T., Zhu, L., & Zhao, P. (2022). Transport inequities through the lens of environmental racism: rural-urban migrants under Covid-19. Transport policy, 122, 26-38.

Sebhatu, A., Wennberg, K., Arora-Jonsson, S., & Lindberg, S. I. (2020). Explaining the homogeneous diffusion of COVID-19 nonpharmaceutical interventions across heterogeneous countries. Proceedings of the National Academy of Sciences, 117(35), 21201-21208.

Wang, X. (2012). Rethinking Universalism in the Context of China. Socialism and Democracy, 26(1), 18-35.

Round 2

Reviewer 2 Report

Thanks for revising. I am satisfied with the current manuscript. It's interesting and informative. A very good paper indeed. Thank you for sharing.